# Omicron vs. the rest: Assessing the competitive dynamics and coinfection scenarios of COVID-19 strains on a social network

**Hamed Jabraeilian, Yousef Jamali** ⬤ *

Biomathematics Laboratory, Department of Applied Mathematics, School of Mathematical Science, Tarbiat Modares University, Tehran, Iran

* y.jamali@modares.ac.ir

## Abstract

The rapid spread and evolving nature of COVID-19 variants have raised concerns regarding their competitive dynamics and coinfection scenarios. In this study, we assess the competitive interactions between the Omicron variant and other prominent variants (Alpha, Beta and Delta) on a social network, considering both single infection and coinfection states. Using the SIRS model, we simulate the progression of these variants and analyze their impact on infection rates, mortality and overall disease burden. Our findings demonstrate that the Alpha and Beta strains exhibit comparable contagion levels, with the Alpha strain displaying higher infection and mortality rates. Moreover, the Delta strain emerges as the most prevalent and virulent strain, surpassing the other variants. When introduced alongside the less virulent Omicron strain, the Delta strain results in higher infection and mortality rates. However, the Omicron strain's dominance leads to an overall increase in disease statistics. Remarkably, our study highlights the efficacy of the Omicron variant in supplanting more virulent strains and its potential role in mitigating the spread of infectious diseases. The Omicron strain demonstrates a competitive advantage over the other variants, suggesting its potential to reduce the severity of the disease and alleviate the burden on healthcare systems. These findings underscore the importance of monitoring and understanding the dynamics of COVID-19 variants, as they can inform effective prevention and mitigation strategies, particularly with the emergence of variants that possess a relative advantage in controlling disease transmission.

## Introduction

The understanding of epidemics plays a crucial role in comprehending disease spread models and developing effective strategies to prevent the over-infection of populations. In this context, the significance of mathematical models cannot be overlooked in the field of epidemiology. The virulence of a pathogen refers to the mortality rate caused by the disease [1]. Virulence and disease transmissibility are key factors in the investigation of disease models.

Multi-infection models encompass the simultaneous spread of several independent viruses within a community. The various types of multiple infections include single infection,

**Data Availability Statement:** All relevant data are within the paper and its Supporting information files.

**Funding:** The authors received no specific funding for this work.

**Competing interests:** The authors have declared that no competing interests exist.

coinfection and superinfection. In the case of single infection, when a strain spreads independently among a population of susceptible individuals, an infected person will only harbor that specific strain until recovery, without contracting another strain. This scenario is termed single infection. Superinfection models account for the possibility of an infected individual being susceptible to multiple strains of the disease simultaneously, with a more virulent strain potentially replacing a less virulent strain through a process akin to hostile takeover [2, 3]. Coinfection models involve individuals sustaining infections with multiple pathogen strains concurrently [4, 5], although typically, susceptible individuals do not contract more than one infection at a time.

In recent years, the COVID-19 epidemic has witnessed the emergence of various variants [6] as reported by the World Health Organization (WHO). A change in the genetic sequence of a virus is referred to as a mutation and variants are genomes that differ from each other in terms of their genetic sequence, often resulting from one or more mutations [7]. As a single-stranded RNA virus [8], SARS-CoV-2 has undergone numerous mutations. While most mutations do not significantly impact the virus's spread and mortality, several mutations have raised global concerns. Hence, it is crucial to develop a better understanding of the transmission of these new coronavirus variants and effective methods to mitigate their spread [9, 10]. Compared to the original lineage, new coronavirus variants exhibit higher transmissibility and increased resistance to antibodies [11].

To prioritize surveillance and research on these variants, the WHO has categorized COVID-19 strains into three groups: variants of interest (VOI), variants of concern (VOC) and variants under monitoring (VUM). The classification of variants may vary across different countries [12]. The four previous VOCs include Alpha (B.1.1.7), Beta (B.1.351), Gamma (P.1) and Delta (B.1.617.2). All of these variants have triggered new waves of epidemics worldwide. On November 26, 2021, the WHO designated a new variant called Omicron (B.1.1.529) as the fifth VOC, instantly sparking global concerns [6]. The term "Wild type" refers to a virus or background strain that does not possess any major mutations [13]. In other words, it represents the natural, non-mutated strain of the virus [14]. The transmission and virulence rates for the main strain, referred to as the Wild type, are based on reference [15].

The main concerns regarding SARS-CoV-2 VOCs encompass viral transmission, disease severity and their effects on vaccine efficacy [16]. The Wild type virus of COVID-19 is more susceptible to neutralization compared to newer variants [17]. According to a study, the Alpha strain exhibited a 43% to 90% higher transmissibility than the Wild type [18]. Additionally, the Alpha lineage demonstrated a 71% higher transmission rate compared to the original lineage [19]. Generally, the Delta and Omicron variants are more transmissible than the Alpha, Beta and Gamma variants [17].

Quantitatively speaking, the Delta variant posed a 108% increased risk of hospitalization, a 235% increased risk of ICU admission and a 133% higher risk of death compared to the original variant [20]. Another study reported that the Delta strain was 60% more transmissible than the Alpha strain, becoming the dominant strain as of August 2021 [21]. Analysis by Bolze et al. revealed that the Delta variant exhibited, on average, 1.7 times higher viral load compared to the Alpha variant [22]. Furthermore, according to another report, the transmissibility of the Delta strain is usually 60% higher than that of the Alpha strain, making it the most infectious variant to date and 97% more contagious than the original strain [23]. Similarly, the Beta strain is 50% more transmissible than the Wild type [24]. Data also indicate that the Delta variant has a higher transmission rate than the Gamma variant [25], with the estimated transmission capability of the Gamma variant being 2.6 times higher than that of the Wild type, according to reference [26]. A report from Public Health England on June 11 indicated a significantly higher risk of hospitalization associated with the Delta variant compared to the Alpha variant [27]. In short, the Delta strain exhibits highly transmissible characteristics and greater invasiveness

[28]. Another study highlighted that the infection rates of the Omicron variant were four times higher than the Wild type and twice as high as the Delta variant [29]. Reference [30] estimates that the Omicron variant is 36.5% more transmissible than the Delta variant. Moreover, a study conducted in Southern California demonstrated that the Omicron strain had a 91% lower fatality rate than the Delta strain and a 51% reduced likelihood of hospitalization [31].

All the VOCs, including Alpha, Beta, Gamma and Delta variants present higher risks of hospitalization, ICU admission and mortality compared to the Wild type virus. Beta and Delta variants pose a higher risk than Alpha and Gamma variants [16]. Alpha, Beta, Gamma and Delta variants had 1.7, 3.6, 2.6 and 2.08 times increased risk of hospitalization, respectively and 2.3, 3.3, 2.2 and 3.35 times increased risk of ICU admission, respectively [16, 32]. These variants also exhibit mortality risks of 1.37, 1.50, 1.06 and 2.33, respectively [16]. The mortality rate of the Alpha strain, as referenced in [33], is reported as 0.005. Qualitatively estimating the transmission and virulence rates for this variant aligns with this reference. Fig 1 illustrates the outcome of our research aimed at investigating the factors influencing the infection, transmission and virulence of COVID-19 variants. All values have been qualitatively investigated, collected and estimated, thus the disease transmission and virulence rates depicted in Fig 1 will be referred to as qualitative rates.

Newer VOCs have largely displaced other circulating SARS-CoV-2 variants. Delta accounted for nearly 90% of all viral sequences submitted to GISAID by October 2021 and Omicron has now become the dominant strain circulating worldwide, representing more than 98% of shared viral sequences in GISAID since February 2022 [34].

The simultaneous circulation of multiple variants in the same location can lead to coinfection with different strains of SARS-CoV-2. Yaqing He et al. reported the identification of an individual co-infected with two worrisome SARS-CoV-2 variants, Beta and Delta [35]. Additionally, Deltacron, involving both Delta and Omicron variants, was first detected in January 2022 and rapidly spread in Cyprus [36]. Several recombination events between the main sub-variants of Omicron (BA.1 and BA.2) and other variants of concern (VOCs) and variants of interest (VOIs) have been observed [37].

Mathematical models are essential tools for studying the spread of COVID-19 in social networks. Manotosh Mandal et al. introduced the SEQIR model, which considers susceptible (S),

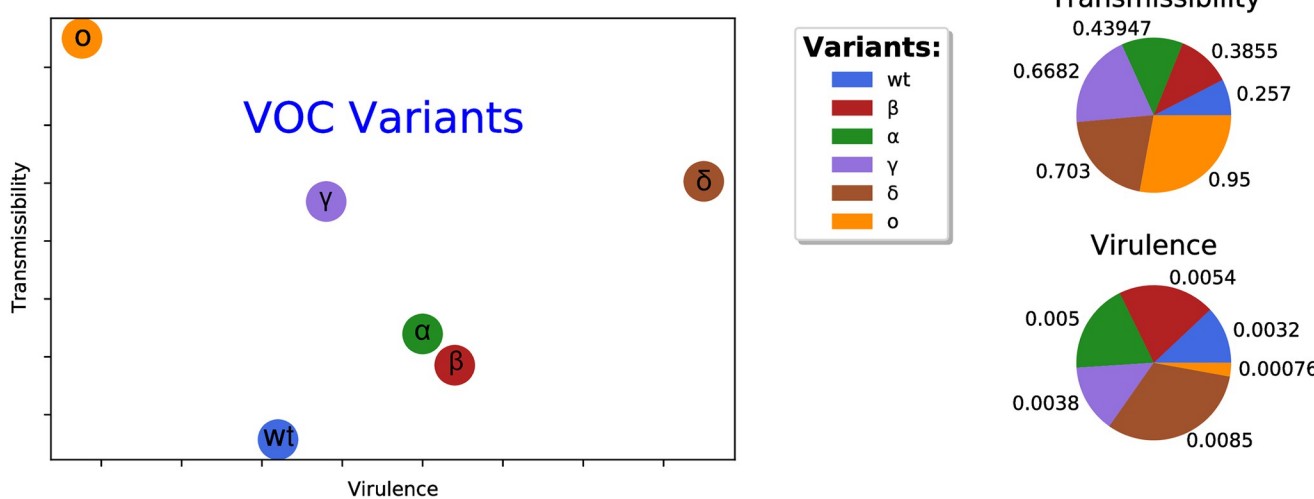

**Fig 1. Qualitative plot of different VOC variants of the disease of Covid-19.**

exposed (E), infected in hospital (Q), quarantined (I) and recovered or eliminated (R) individuals [38]. Joe Pharaon and Chris T compared the dynamics of a two-strain SIRX compartmental epidemic model with and without adaptive social behavior, where susceptible individuals (S) can be infected by a resident strain ($I_1$) or a mutant strain ($I_2$). The proportion of individuals adopting preventive behaviors is represented by X, referred to as protective behavior [39]. Cleo Anastassopoulou et al. proposed the SIRD model to estimate parameters and predict the spread of the COVID-19 epidemic in Hubei Province, China, using real data [40]. Verónica Miró Pina et al. developed models based on Erdős-Rényi and a power-law degree distribution, which capture the role of heterogeneity and connectivity and can be extended to make hypotheses about demographic characteristics. They demonstrated that changes in the number of contacts in a population impact the effectiveness of public health interventions such as quarantine or vaccination strategies [41]. Pilar Hernández et al. utilized the SEIR model and presented two types of agent-based model simulations: a homogeneous spatial simulation where contamination occurs through proximity and a model in a scale-free network with different clustering characteristics, where contamination occurs between any two agents through their link, if any [42]. Mosquera and Adler established consistency conditions on the parameters of a mathematical model for coinfection states, including superinfection as a limit of the coinfection model and a special single infection state [43].

In this manuscript, we present the SIRS model as an extension of the framework developed by Mosquera and Adler. Our study entails a comprehensive analysis and discussion of simulations conducted on a social network, incorporating both single infection and coinfection states. With a specific emphasis on the emerging variants of COVID-19, we explore the intricate dynamics of strain competition within the contexts of single infection, coinfection and delayed states. Furthermore, we thoroughly investigate the influence of network topology on the overall progression of the epidemic.

## Model

We investigate the scenario of multiple infections by considering a simplified case where only one or two infectious diseases can occur concurrently within the population. In the context of the SIRS model applied to the network, we mathematically describe the dynamics of the epidemic. Consequently, the network consists of four distinct agent types: susceptible individuals (S), individuals infected with virus strain 1 ($I_1$), individuals infected with virus strain 2 ($I_2$) and individuals simultaneously infected with both strains ($I_{12}$). Notably, the susceptible class is further categorized into two groups: susceptible individuals 1 ($S_1$) who have not been infected and are susceptible to receiving the infection and susceptible individuals 2 ($S_2$) who have previously experienced an infectious disease but remain susceptible with a lower probability.

Examining the impact of mask usage on COVID-19 transmission, a study directly analyzed its effect within the community. The findings revealed that if face masks were universally worn within a household before symptom onset, they effectively reduced transmission by 79% [44]. Masks play a crucial role in preventing the spread of droplets and aerosols emitted by infected individuals [45]. Correct usage of surgical masks can significantly decrease virus transmission by approximately 95% providing around 85% protection against infection for non-infected individuals [46]. Furthermore, another study identified mental health status and social media as factors influencing adherence to social distancing measures. The high compliance rate of social distancing measures was reported by the majority of respondents (95.6%) [47]. Thus, assuming adherence to health protocols such as mask-wearing, social distancing and other preventive measures against COVID-19 transmission, we define the protection parameter (p)

as the reduction in the probability of infection transmission. In light of the aforementioned cases, we set the protection parameter to p = 0.9.

## Coinfection model

In order to define the coinfection model, we make the following assumptions:

1. The mortality rate of individuals due to non-disease factors during the simulation interval can be neglected.

2. Infected individuals with strain i recover at a rate of $\gamma_i$ and have a mortality rate of $\delta_i$ attributed to the infection. They are then classified as removed (R).

3. Individuals infected with both strains simultaneously experience a disease-related mortality rate of $\delta_{12}$. These individuals recover from strain 1 viral disease at a rate of $\gamma_{21}$, while coinfected individuals recover from strain 2 viral disease at a rate of $\gamma_{12}$.

4. Susceptible individuals become infected by individuals infected with strains 1 and 2 at rates of $\beta_1$ and $\beta_2$, respectively.

5. The transmissibility rate for individuals co-infected with both strains can differ from the rate for individuals infected with strains 1 and 2 separately. Specifically, if the reduced infectiousness of co-infected individuals acting as strain i is denoted by $\varepsilon_i$, the rate at which susceptible individuals acquire strain i from co-infected individuals will be $\varepsilon_i \beta_i I_{12}$, where $\varepsilon_i$ belongs to the range [0, 1]. This means that when a susceptible individual interacts with a co-infected individual, the transmission of either $I_1$ or $I_2$ disease (depending on the competition between strains) occurs at a lower rate.

6. If there is reduced susceptibility to the other strain when infected with strain i, denoted by $a_i$, the infection rate of $I_1$ individuals by strain 2 is less than the infection rate of susceptible individuals by a factor of $a_2$. Similarly, the infection rate of $I_2$ individuals by strain 1 is equal to $a_1 \beta_1 I_1 + a_1 \varepsilon_1 \beta_1 I_{12}$.

7. In the case of reinfection, susceptible individuals 2 ($S_2$) contract strain i at a much lower transmission rate of $\alpha \beta_i$.

Based on these assumptions, the coinfection model follows the set of equations below, where the total number of individuals in the network is denoted as N = $S_1 + S_2 + I_1 + I_2 + I_{12} + R$:

$$\frac{dS_1}{dt} = -(1-p)\{\beta_1 I_1 S_1 + \beta_2 I_2 S_1 + \varepsilon_1 \beta_1 I_{12} S_1 + \varepsilon_2 \beta_2 I_{12} S_1\}$$

$$\frac{dS_2}{dt} = -\alpha(1-p)\{\beta_1 I_1 S_2 + \beta_2 I_2 S_2 + \varepsilon_1 \beta_1 I_{12} S_2 + \varepsilon_2 \beta_2 I_{12} S_2\} + \gamma_1 I_1 + \gamma_2 I_2$$

$$\frac{dI_1}{dt} = (1-p)\beta_1 (S_1 + \alpha S_2)(I_1 + \varepsilon_1 I_{12}) - (\delta_1 + \gamma_1)I_1 - a_2 \beta_2 I_1 I_2 + \gamma_{12} I_{12} - \varepsilon_2 a_2 \beta_2 I_1 I_{12}$$

$$\frac{dI_2}{dt} = (1-p)\beta_2 (S_1 + \alpha S_2)(I_2 + \varepsilon_2 I_{12}) - (\delta_2 + \gamma_2)I_2 - a_1 \beta_1 I_1 I_2 + \gamma_{21} I_{12} - \varepsilon_1 a_1 \beta_1 I_2 I_{12} \qquad (1)$$

$$\frac{dI_{12}}{dt} = (a_1 \beta_1 + a_2 \beta_2)I_1 I_2 + (\varepsilon_2 a_2 \beta_2 I_1 + \varepsilon_1 a_1 \beta_1 I_2)I_{12} - (\gamma_{12} + \gamma_{21} + \delta_{12})I_{12}$$

$$\frac{dR}{dt} = \delta_1 I_1 + \delta_2 I_2 + \delta_{12} I_{12}$$

Generally, Fig 2 illustrates the set of Eq (1) that represent the coinfection model:

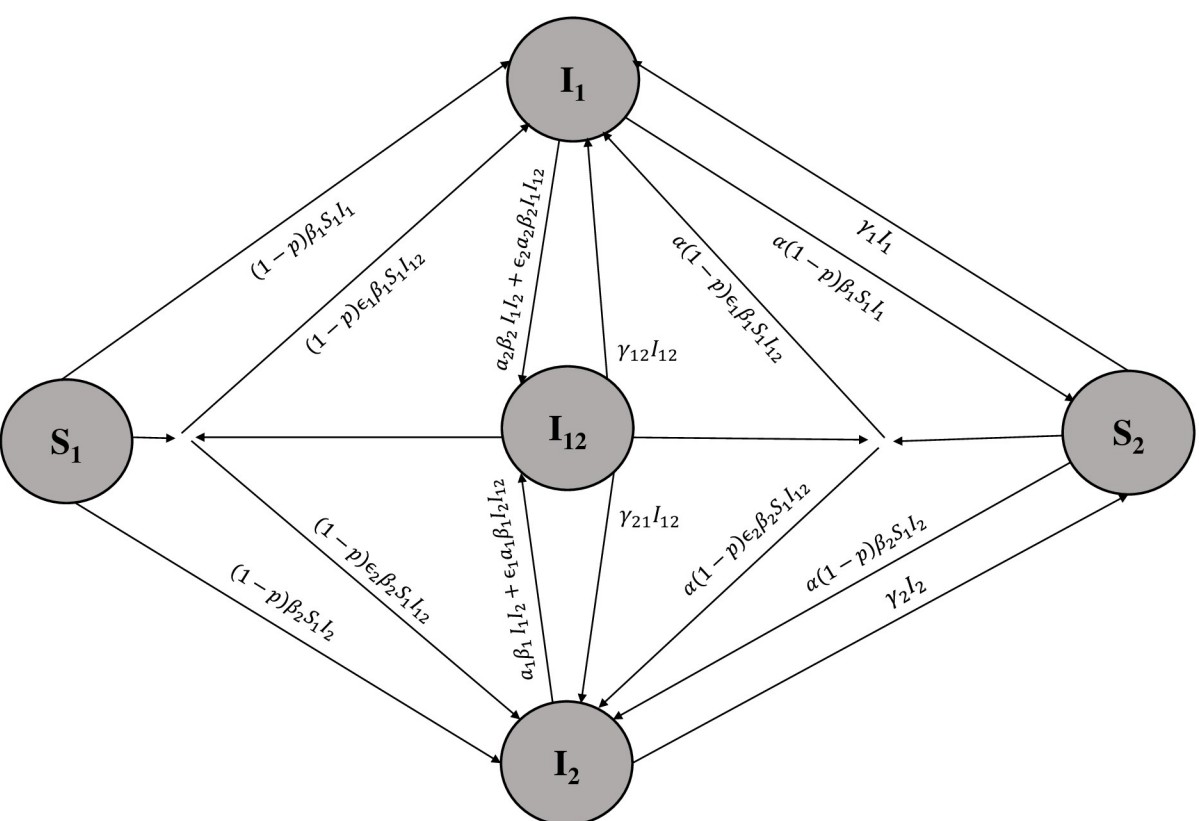

**Fig 2. Schematic diagram illustrating the interaction between two different strains in the coinfection model.** Susceptible individuals $S_1$ can become infected by interacting with infectious individuals $I_1$ and $I_{12}$ at rates $\beta_1$ and $\varepsilon_1\beta_1$, respectively. If they have protection, they are transferred to the $I_1$ class with a coefficient of $(1\text{-}p)$. Infectious individuals $I_1$ can transition to the $I_{12}$ class by interacting with infectious individuals $I_2$ and $I_{12}$ at rates $a_2\beta_2$ and $\varepsilon_2\, a_2\beta_2$, respectively. Individuals in the $I_1$ class are removed at a rate of $\delta_1$ and recover at a rate of $\gamma_1$, transitioning to the $S_2$ class. The process is similar for susceptible individuals $S_2$, but at a much lower rate. The same interactions and transitions apply to individuals in the $I_2$ class as well.

### Single infection model

In the single infection model, infected individuals are not susceptible to further infection. By setting the coinfection coefficient equal to zero ($a_1 = a_2 = 0$), which implies that $I_{12} = 0$, the dynamic terms of the coinfection model are simplified, resulting in the follow- ing model:

$$\frac{dS_1}{dt} = -(1-p)\{\beta_1 I_1 S_1 + \beta_2 I_2 S_1\}$$

$$\frac{dS_2}{dt} = -\alpha(1-p)\{\beta_1 I_1 S_2 + \beta_2 I_2 S_2\} + \gamma_1 I_1 + \gamma_2 I_2$$

$$\frac{dI_1}{dt} = (1-p)\beta_1(S_1 + \alpha S_2)I_1 - (\delta_1 + \gamma_1)I_1 \qquad (2)$$

$$\frac{dI_2}{dt} = (1-p)\beta_2(S_1 + \alpha S_2)I_2 - (\delta_2 + \gamma_2)I_2$$

$$\frac{dR}{dt} = \delta_1 I_1 + \delta_2 I_2$$

**Table 1. Values of parameters of various models of Covid-19.**

| Models | $\beta$ | $\gamma$ | $\delta$ | References |
|---|---|---|---|---|
| SIR | 0.4 | 0.2 | 0.05 | [39] |
| SIRD | 0.21542 | 0.017129 | 0.011832 | [48] |
| SIRD | 0.257 | 0.0315 | 0.0032 | [15] |
| SAIU | 0.274 | - | - | [49] |
| SIQR | 0.13 | 0.15 | 0.038 | [50] |
| SEIR | - | - | 0.0018 | [51] |
| SEIR | 0.29 | 0.09722 | - | [52] |
| SEAIR | - | 0.13978 | 0.015 | [53] |
| SEIAHRD | 0.38974 | 0.1428 | 0.015 | [54] |
| SEAIQHR | 1.11525 | 0.01496 | 0.04142 | [55] |

For the model implementation, we adopt an Erdős–Rényi random network with a total of N = 10,000 nodes and an average degree of k = 10. The simulation is conducted for a specified time interval T and each run is repeated 50 times for reliable results.

To examine the competition and coinfection dynamics between different strains, we consider the rates associated with the Covid-19 epidemic. It is important to note that in each simulation, the presence of two strain types in the community is taken into account. Table 1 presents the parameter values for various models of Covid-19, which are derived from ten different references [15, 39, 48–55]. Notably, the units in the table are given in terms of per day ($day^{-1}$). To incorporate real-world data, we refer to reference [15] for the parameters related to the Wild type of the virus.

## Results and discussion

In our study, we focused on the variants of Covid-19, specifically the Alpha, Beta, Delta and Omicron strains. By utilizing the findings of our research, we conducted simulations to examine the dynamics of these strains.

### Investigating the competition between covid-19 variants with strain rates

We began by considering the single infection state for the four VOC strains and observed their dynamic changes during the epidemic, as shown in Fig 3. The contagion of the Alpha and Beta strains was relatively similar, with the Alpha strain exhibiting approximately 5% more cases than the Beta strain. However, compared to the Wild type, the Alpha strain had a peak infection and mortality rate almost twice as high. Furthermore, the Delta strain exhibited a higher prevalence of disease and mortality compared to the other three strains. We also examined the competition between strains in the following scenarios:

1. Competition between the Wild type and the Alpha variant

2. Competition between the Alpha and the Beta variants

3. Competition between the Delta and the Omicron variants

Initially, we evaluated the Wild type and the Alpha mutant variant in a society with a size of N = 10,000. As depicted in Fig 4a, the number of infected individuals for the main strain and the Alpha strain reached their maximum values after t = 59 days ($I_{1max}$=109) and t = 73 days ($I_{2max}$=2199), respectively. By t = 200 days, the number of infected individuals for the main

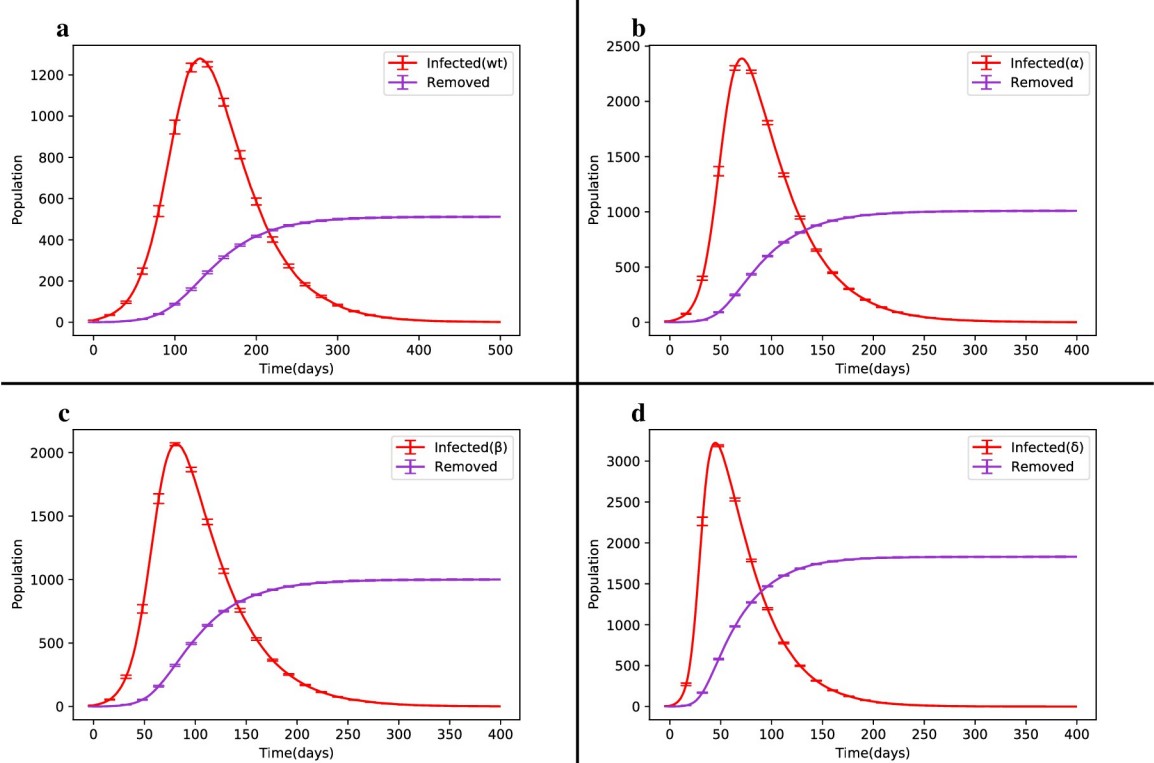

**Fig 3. Dynamic changes of the SIRS model in the single infection state.** (a) Outbreak of the Wild type. (b) Outbreak of the Alpha variant. (c) Outbreak of the Beta variant. (d) Outbreak of the Delta variant in the Erdős–Rényi network.

strain decreased to 4, while for the Alpha strain, it was 167. It is worth noting that the mutated Alpha strain had a wider epidemic range and resulted in a significantly higher number of infections compared to the original strain. The number of individuals who experienced either the main strain or the Alpha strain at least once was 3.21% and 68.50%, respectively.

In Fig 4b, the total number of Covid-19 cases through the Alpha strain was nearly 2.8 times higher than that of the Beta strain. The key difference between Fig 4a and 4b is that the Beta strain replaced the main strain and the overall number of infected individuals was distributed in a ratio of 3 to 1. An important concern arises when two highly transmissible strains enter the community. Thus, we introduced the more virulent Delta variant and the less virulent Omicron variant into the network and their interaction among nodes yielded the dynamics of infected individuals as shown in Fig 4c.

The total number of infections for the Delta and Omicron strains was 11.54% and 89.96%, respectively. The mortality rate due to the disease was higher for the Delta strain, as its virulence was 91% greater than that of the Omicron strain. Even though the recovery rates were assumed to be the same for both strains, the number of deaths through the Delta and Omicron strains was 24.3% and 20.8%, respectively. It is important to note that the majority of deaths occurred as a result of the Delta strain.

If we consider the time to reach the peak of the disease ($t_{max}$), we observe that it is almost half the time in Fig 4c compared to Fig 4a and 4b. This is the reason why these strains are of concern to the World Health Organization (WHO). Due to their higher prevalence, a greater number of susceptible individuals are affected by the disease in a shorter period of time,

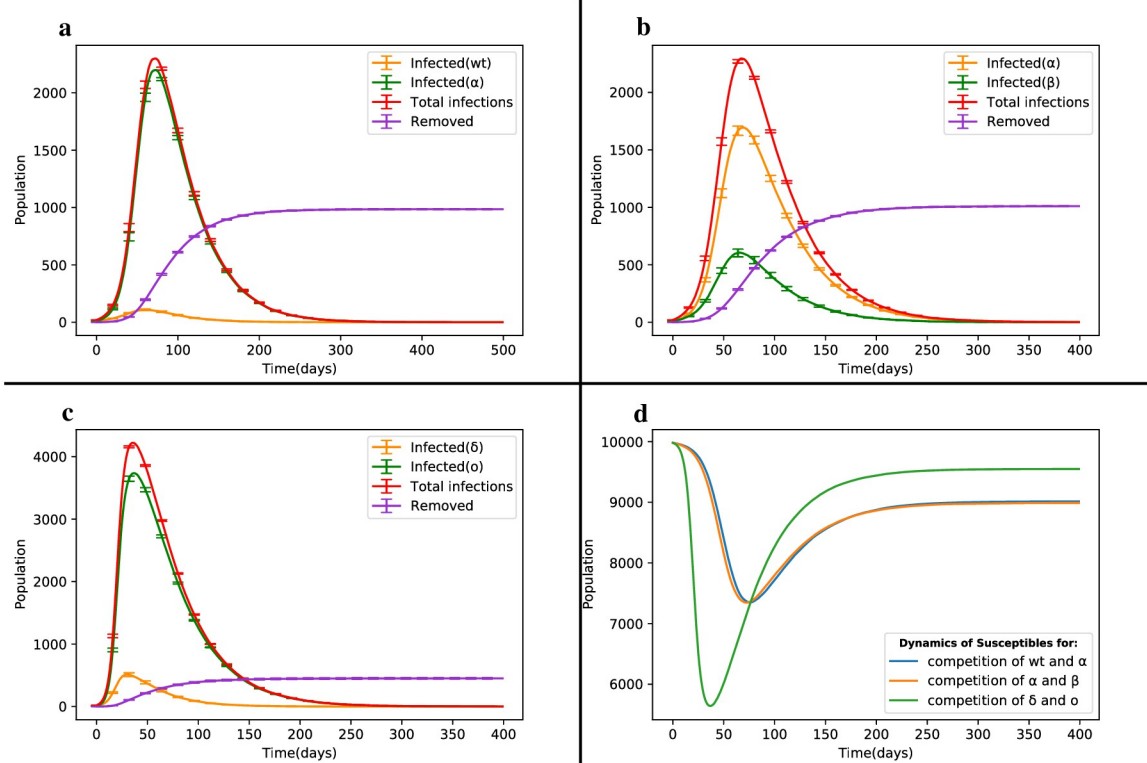

**Fig 4. Dynamic changes of the SIRS model in the competition between strains.** (a) Competition between the Wild type and Alpha mutant strain. (b) Competition between Alpha and Beta strains. (c) Competition between Delta and Omicron strains. (d) Dynamics changes of susceptible individuals.

resulting in a 28.33% increase in the number of infected individuals compared to the previous scenario. The dynamic curve of susceptible individuals in the competition between wt and $\alpha$ strains with $\alpha$ and $\beta$ strains is similar. However, as depicted in Fig 4d, when competing Delta and Omicron strains are involved, a larger number of susceptible individuals are affected by the disease. The difference in the minimum points between the previous two cases and the third case is 17.04%. This suggests that the virulence of both strains was higher in the previous two cases, but over time, the equilibrium point of susceptible individuals in the third case is 5.6% lower than that in the other two cases. Table 2 provides detailed information on these three cases for the competition of Covid-19 strains.

**Table 2. Statistics of the results obtained for the SIRS model, in the state of competition different strains of Covid-19.**

| Quantity | Description | State1 | State2 | State3 |
|---|---|---|---|---|
| $I_{1max}$ | Peak domain of dynamics of infected individuals to strain 1 | 109 | 1692 | 513 |
| $I_{2max}$ | Peak domain of dynamics of infected individuals to strain 2 | 2199 | 606 | 3735 |
| $I_{1avg}$ | The total mean of dynamics of infected individuals to strain 1 | 19 | 365 | 70 |
| $I_{2avg}$ | The total mean of dynamics of infected individuals to strain 2 | 3790 | 126 | 685 |
| $S_{min}$ | The minimum value of dynamics of susceptible individuals at the peak of the disease | 7346 | 7345 | 5642 |
| $TotalInfected$ | The total stats of infected individuals to disease | 7369 | 7317 | 10150 |
| $TotalRecovered$ | The total stats of recovered individuals from disease | 6384 | 6306 | 9698 |
| $TotalRemovedduetoinfection$ | The total stats of removed individuals from disease | 985 | 1010 | 451 |

Based on the findings from Figs 3 and 4, we can conclude that even if the transmissibility of a strain is approximately 70% higher than the other strain, most of the disease statistics in the society will often be attributed to the strain with the higher transmission rate (as seen in Fig 4a and 4b). Additionally, due to the higher transmission and virulence rates of the Delta strain compared to the Wild type, Alpha and Beta strains, the introduction of the Omicron strain, despite its lower virulence, resulted in more disease statistics compared to all the other strains. Therefore, this situation is a cause for concern in terms of increasing the overall disease burden, but it could also be seen as a positive scenario for the removal of a more dangerous strain from the society.

### The delayed state of competition of Covid-19 strains

In this section, we explore the competition between the Wild type-Omicron and Alpha-Omicron strains in a delayed state. In this scenario, the main strain initially spreads in the network and after a certain period, the second strain is introduced. Fig 3a and 3b depict the outbreak of the disease through the main and Alpha strains, respectively. The Alpha strain exhibits twice the mortality rate and peak infection level compared to the main strain. Our goal is to introduce a strain with a higher transmission rate into the network with a delay.

In Fig 5a, we introduce the less virulent Omicron strain when the main strain is in its maximum growth trend (maximum slope). In Fig 5c, we insert the second strain before the maximum slope of the first strain. As observed in Fig 5a and 5c, the addition of the less virulent Omicron strain, with a higher prevalence compared to the main and Alpha strains (4.6 and 4.8

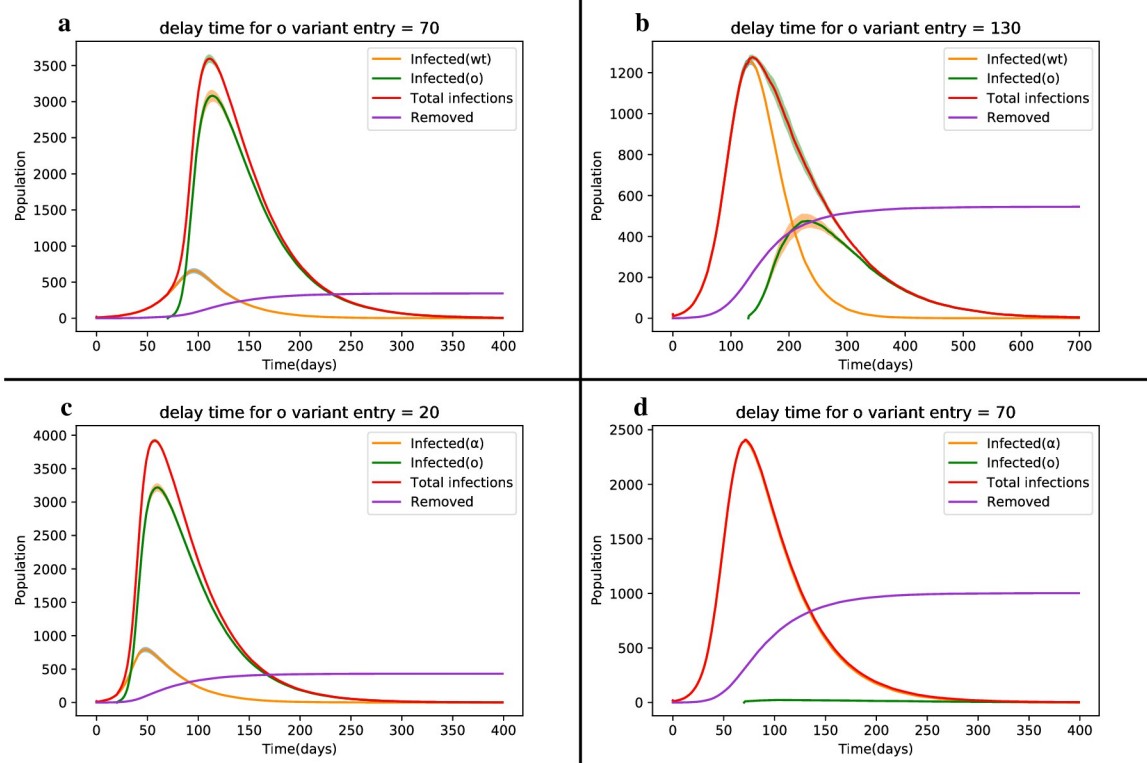

**Fig 5. Delay state diagram depicting the dynamic changes of the SIRS model during the competition between the Wild type, Alpha and Omicron strains.** The colored hachure in the peak of the curves represents the error bars for 50 times experiments in each run.

times, respectively), results in a greater number of disease cases. In other words, the Omicron strain effectively suppresses the more virulent strains, acting as a vaccine-like agent. This process continues until the end of the epidemic. Consequently, as shown in Fig 5, the dynamics of deaths are significantly lower in parts a and c, primarily due to the predominance of the Omicron strain.

It is noteworthy that the second strain must have a higher transmission rate to effectively counteract the more virulent strains. Some experts suggest that Omicron may act as a natural vaccine since it shares certain similarities with weakened live vaccines [56]. Studies have shown that individuals infected with Omicron exhibit a significant immune response that can neutralize not only Omicron but also other variants of concern, including the prevalent Delta variant [57].

If we introduce the second strain during the peak of infected individuals with the Alpha strain, as shown in Fig 5b and 5d, the Omicron strain fails to surpass the previous strain with higher transmissibility. The Wild type and Alpha strain, which are more contagious and fatal than the Omicron strain, result in 1.7 and 55 times more disease cases, respectively. Since the Alpha strain has infected a larger portion of the population, leading to more individuals entering the $S_2$ class, the Omicron strain struggles to infect these individuals effectively. Consequently, the dynamics of infected individuals to the second strain appear almost as a straight line.

Overall, the delayed introduction of the less virulent Omicron strain can significantly reduce the disease burden caused by more virulent strains, highlighting its potential role as a protective factor against the spread of highly contagious variants.

### Investigating coinfection between covid-19 variants

In this section, we examine the coinfection dynamics between the Delta and Omicron strains, collectively referred to as Deltacron, by analyzing the behavior of co-infected individuals for different values of parameter a, specifically a = 0.1, 0.5, 1 and 2.

When considering values of a < 1 (see Fig 6), which correspond to lower levels of coinfection, we observe that the dynamics of infected individuals for the Delta strain reaches its peak 10 days earlier than that of the Omicron strain. This can be attributed to the weaker co-infectious conversion between infected individuals of the Omicron and Delta strains, resulting in a lower rate of conversion and the earlier peak for the Delta strain due to its higher virulence.

There are two noteworthy observations in this scenario. Firstly, the slope reduction between the interval [50, 100] for the Delta strain dynamics is more significant for a = 0.1 compared to a = 0.5. This reduction becomes smoother as the presence of co-infected individuals increases and they transition to classes $I_1$ and $I_2$. For example, the slope decreases from -5.6 for a = 0.1 to -3.8 for a = 0.5 in this interval.

Secondly, as the coefficient a increases, the dynamics of co-infected individuals show an overall increase. Despite the Omicron strain being dominant, the shape of the Delta strain dynamics differs before reaching its peak (see Fig 5c and 5d). For instance, when a = 1 in the interval [10, 20], all three dynamics initially exhibit an increasing trend, with the share of coinfection through the Delta strain being approximately 20% higher than that of Omicron. This is due to the random network structure and the interaction between infected individuals of both strains, where the Delta strain has a higher chance of selection for coinfection. Over time, as the Omicron strain becomes more prevalent, it gradually surpasses the Delta strain, reaching its peak at t = 32. In the interval [20, 32], the share of coinfection through the Omicron strain becomes greater than that of Delta, leading to a gradual decrease in the dynamics of Delta-infected individuals with a slope of -2. Meanwhile, the dynamics of individuals in class $I_1$

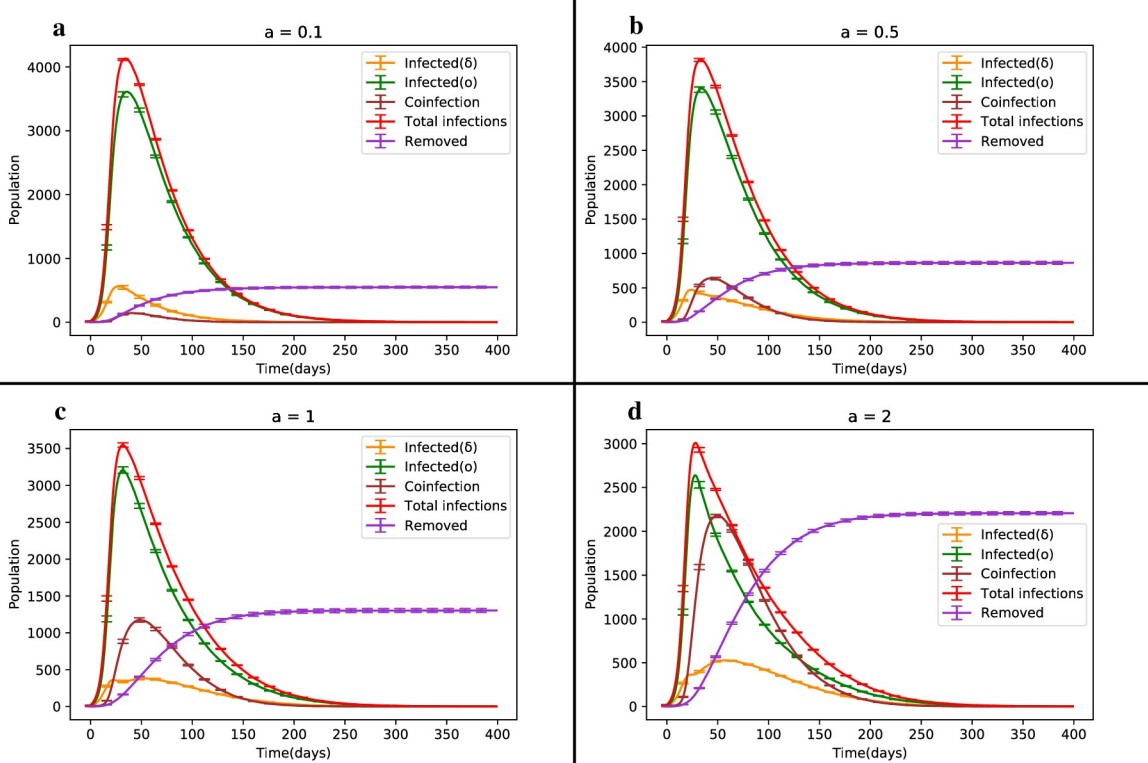

**Fig 6. Dynamic changes diagram of the SIRS model illustrating the competition between different variants of Covid-19 for different coefficients of a in the coinfection state.**

continue to increase until they reach their peak at t = 51. Moreover, in [32, 51], the Omicron strain is involved in coinfection nearly 60% more than the Delta strain. As the dynamics of class $I_2$ decline and the dynamics of class $I_1$ culminate, the dynamics of coinfection align with that of class $I_1$. From t = 51 onwards, all three dynamics exhibit a decreasing trend. Consequently, higher values of a lead to an earlier peak in the dynamics of infected individuals for the Omicron strain compared to the Delta strain and coinfection. Subsequently, all three curves gradually decline, signifying the end of the epidemic after approximately 250 days.

In general, if individuals in the coinfection class ($I_{12}$) are rapidly removed, the $I_{12}$ class decreases to zero. The total number of removals, as indicated by relation (1) [43], is given by Γ = $\gamma_{12} + \gamma_{21} + \delta_{12}$. Thus, the reduction of co-infected individuals is approximated by the rapid elimination of these individuals through rapid recovery from either strain or by an increase in mortality due to double infection itself. Table 3 provides the results for the four aforementioned cases. It is worth noting that although the network size is N = $10^4$, the numbers in Table 3 appear larger. For instance, in the case of a = 2, the total number of patients is 23,459, implying that individuals who have had the disease and recovered remain susceptible to reinfection. For this specific case, the number of individuals infected with the Delta, Omicron and Deltacron strains at least once is 2,203, 7,680 and 4,997, respectively.

With an increase in the coefficient $a_i$, a remarkable rise in the dynamics of the coinfection class is observed. Additionally, the dynamics of susceptible individuals decrease, as illustrated in Fig 7.

**Table 3. Statistics of SIRS model results, for four coinfection state of Deltacron.**

| Quantity | Description | a = 0.1 | a = 0.5 | a = 1 | a = 2 |
|---|---|---|---|---|---|
| $I_{1max}$ | Peak domain of dynamics of infected individuals to strain 1 | 562 | 476 | 385 | 529 |
| $I_{2max}$ | Peak domain of dynamics of infected individuals to strain 2 | 3611 | 3400 | 3208 | 2640 |
| $I_{12max}$ | Peak domain of dynamics of co-infected individuals to strain 12 | 145 | 644 | 1177 | 2181 |
| $I_{1avg}$ | The total mean of dynamics of infected individuals to strain 1 | 80 | 92 | 108 | 159 |
| $I_{2avg}$ | The total mean of dynamics of infected individuals to strain 2 | 672 | 642 | 598 | 487 |
| $I_{12avg}$ | The total mean of dynamics of co-infected individuals to strain 12 | 20 | 104 | 218 | 445 |
| $S_{min}$ | The minimum value of dynamics of susceptible individuals at the peak of the disease | 5569 | 5390 | 5232 | 4777 |
| TotalInfected | The total stats of infected individuals to disease | 10744 | 13294 | 16737 | 23459 |
| TotalRecovered | The total stats of recovered individuals from disease | 10137 | 12083 | 14694 | 19779 |
| TotalRemovedduetoinfection | The total stats of removed individuals from disease | 5490 | 864 | 1304 | 2206 |

## The impact of topology

This section explores the influence of network topology on the dynamics of infection spreading. In addition to the previously discussed Erdős-Rényi (ER) network, we now investigate the Barabási-Albert (BA) and Watts-Strogatz (WS) networks to understand the competition between the Omicron and Delta variants, as well as the coinfection state involving these two strains.

We examine the performance of the model on the WS network with different rewiring probabilities: P = 0 (representing the regular state), P = 0.1 (exhibiting small-world features) and P = 1 (representing the random state). This analysis encompasses both the single infection and coinfection models.

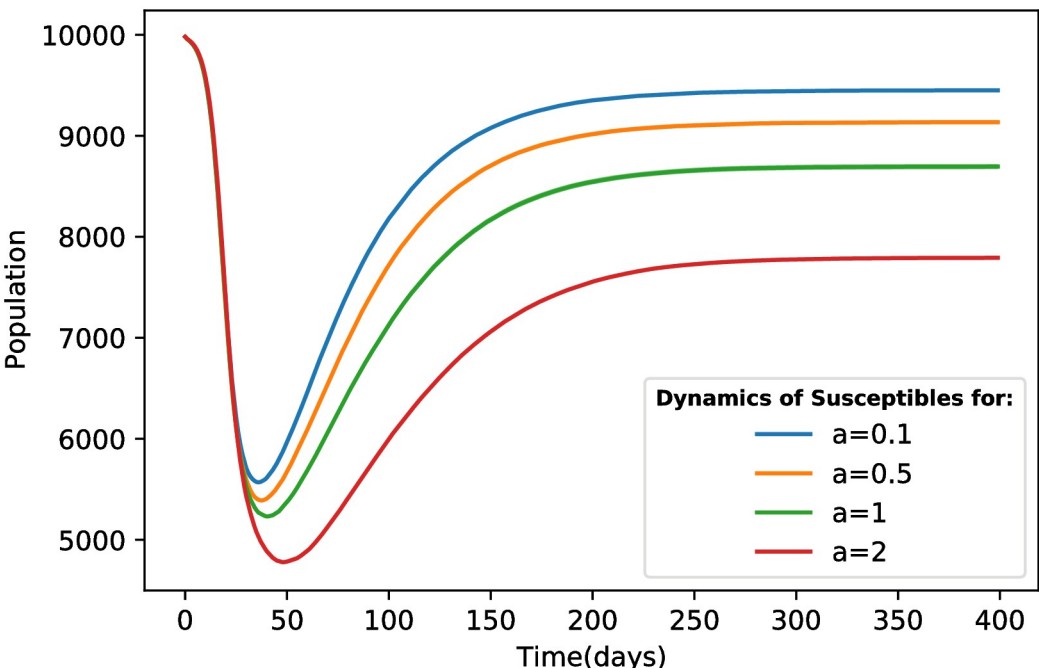

**Fig 7. Dynamic changes of susceptible individuals in four different scenarios of Deltacron outbreak.**

Despite the heterogeneity and the presence of a hub in the BA network (due to the limited network size), the results associated with this network structure align with those observed in the ER random network and the WS random state. Fig 8 illustrates that no significant differences are discernible, with the average total disease proportion in parts c and d being equal to 7.55%. The average ratio of the clustering coefficient to the average path length in parts a, b and c is 0.001, 0.08 and 0.0002, respectively. Furthermore, the average total disease proportions for cases a, b and c are 0.64%, 7.47% and 7.55%, respectively. The larger average path length in case a contributes to a mortality rate that is 3.2 times higher than that observed in the other two cases. By comparing these three states, we find that the regular state of the WS network exhibits a smaller disease prevalence throughout the epidemic.

The primary distinction between states b and c lies in the infection peak time for each strain. Although the curves in these two states are almost identical, the small-world feature causes infections to occur later. The formation of clusters in the network leads to a higher prevalence of single-strain infections compared to coinfections. Conversely, in the random state, the occurrence of two-strain infections is more widespread, resulting in earlier infections among nodes. As depicted in Fig 9, the difference in the trough point of susceptible individuals dynamics for part b compared to the WS random network is 5.38% and this drop occurs 22 days later than in the random state.

All three states b, c and d exhibit similar disease prevalence. Therefore, at the end of the epidemic, they converge to the same equilibrium point in Fig 9. Compared to the regular case, the difference in the equilibrium state is approximately 3%, indicating that the number of removed

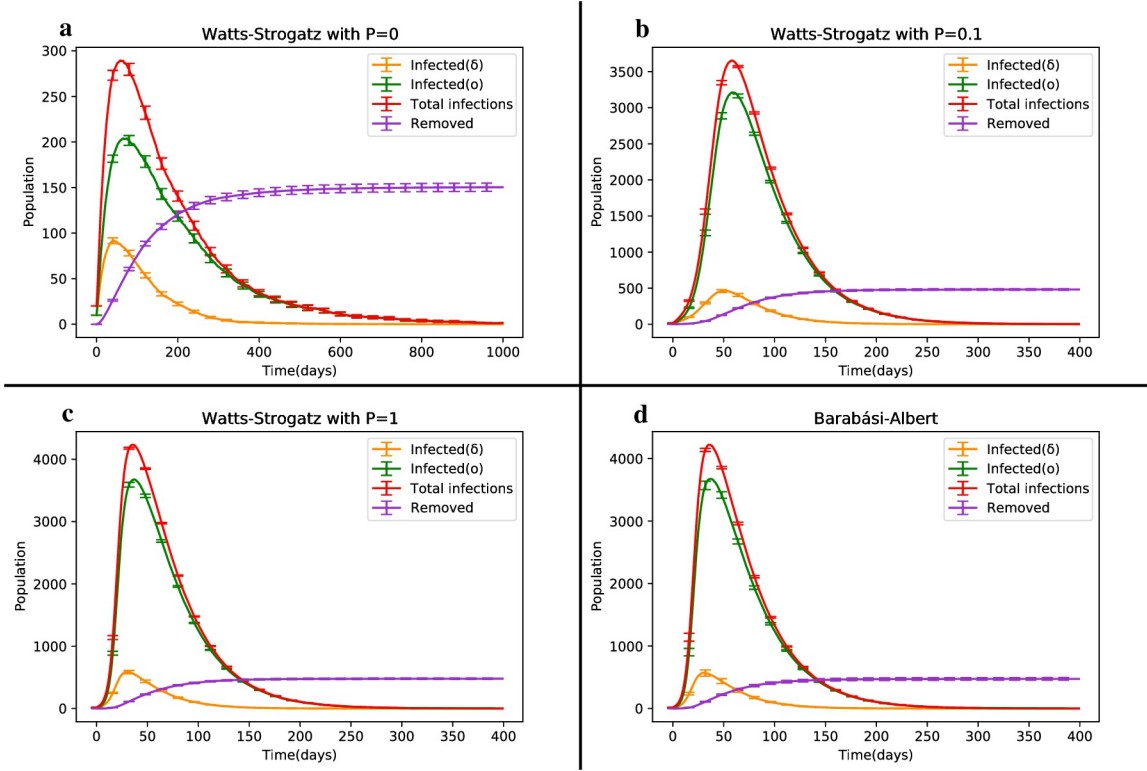

**Fig 8. Dynamic changes of the SIRS model depicting the competition between Delta and Omicron strains in the Barabasi-Albert (BA) and Watts-Strogatz (WS) network structures.**

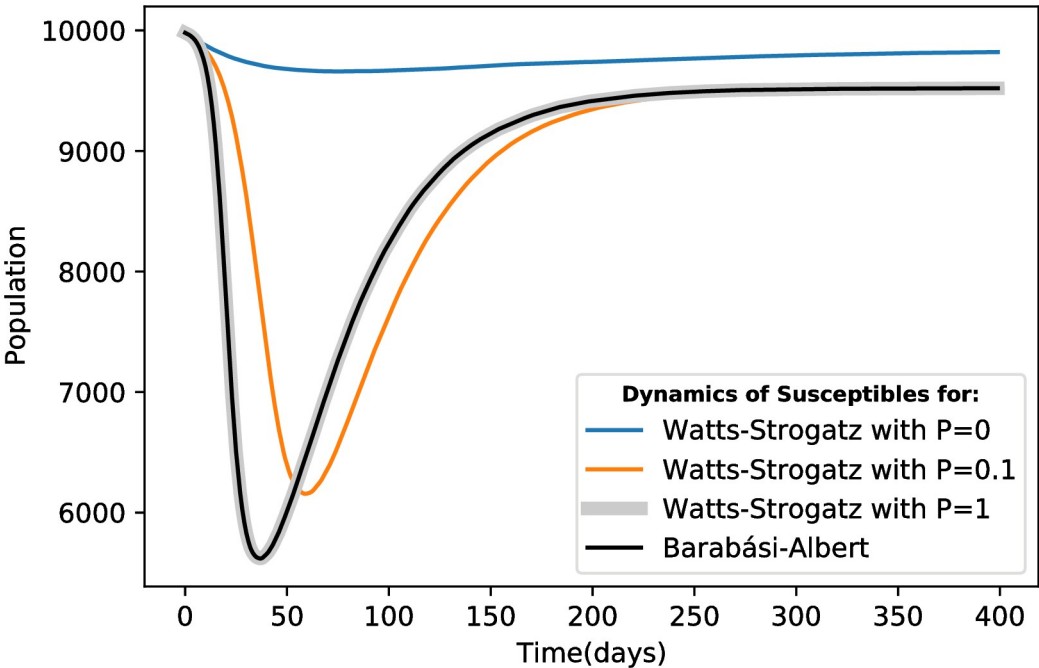

**Fig 9. Dynamic changes of susceptible individuals in the Watts-Strogatz (WS) and Barabasi-Albert (BA) network structures for three different states.**

individuals is around 300 higher in the random network than in the regular network (refer to Table 4).

The coinfection state of the four aforementioned network structures is depicted in Fig 10. It is evident that the average number of dynamically co-infected individuals in the WS network with P = 0 is negligible compared to the other structures. In part b (representing the small-world network), the average total disease prevalence is approximately 0.2% higher than in the two random cases, c and d. However, the average coinfection rate in part b is only half of that observed in the random cases. This reduced occurrence of strain coexistence represents the main difference between part b and the random states, c and d (refer to Table 5).

**Table 4. Summary statistics from the SIRS model showcasing the competition between various strains of Covid-19 across different network structures.**

| Quantity | Description | WS(P = 0) | WS(P = 0.1) | WS(P = 1) | ER | BA |
|---|---|---|---|---|---|---|
| $I_{1max}$ | Peak domain of dynamics of infected individuals to strain 1 | 91 | 463 | 587 | 513 | 571 |
| $I_{2max}$ | Peak domain of dynamics of infected individuals to strain 2 | 204 | 3212 | 3677 | 3735 | 3674 |
| $I_{1avg}$ | The total mean of dynamics of infected individuals to strain1 | 13 | 81 | 80 | 70 | 79 |
| $I_{2avg}$ | The total mean of dynamics of infected individuals to strain 2 | 51 | 665 | 676 | 685 | 671 |
| $S_{min}$ | The minimum value of dynamics of susceptible individuals at the peak of the disease | 9660 | 6156 | 5618 | 5642 | 5637 |
| $t_{max}(\delta)$ | The time of culminated dynamics of infected individuals to Delta strain | 40 | 51 | 30 | 30 | 30 |
| $t_{max}(o)$ | The time of culminated dynamics of infected individuals to Omicron strain | 66 | 59 | 37 | 37 | 37 |
| *TotalInfected* | The total stats of infected individuals to disease | 2175 | 10049 | 10156 | 10150 | 10097 |
| *TotalRecovered* | The total stats of recovered individuals from disease | 2025 | 9565 | 9677 | 9698 | 9622 |
| *TotalRemovedduetoinfection* | The total stats of removed individuals from disease | 150 | 482 | 479 | 451 | 474 |

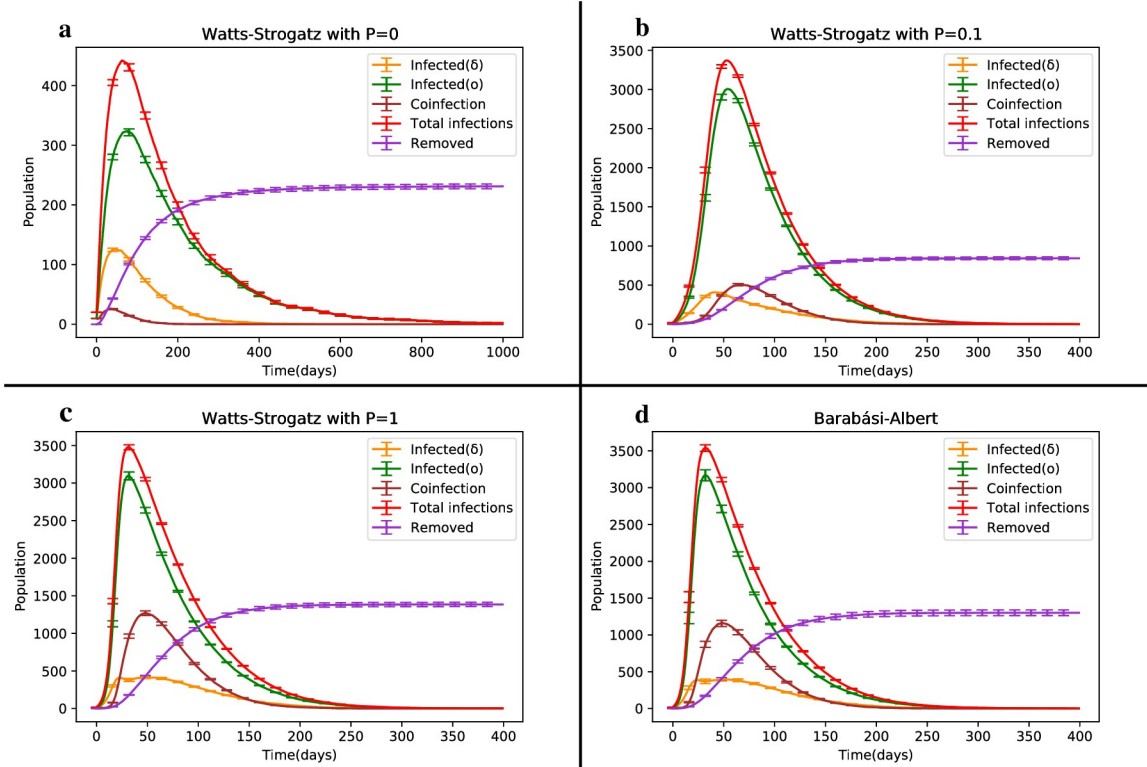

**Fig 10. Temporal evolution of the SIRS model in the coinfection state of Delta and Omicron strains, comparing the network structures of Barabasi-Albert (BA) and Watts-Strogatz (WS).**

As shown in Fig 10b, 10c and 10d, the mortality caused by Deltacron is higher than that caused by both the Delta and Omicron strains. Additionally, the mortality resulting from coinfection in part b is roughly half of that observed in the random states. Recent explanations indicate that Deltacron exhibits significantly higher virulence.

**Table 5. Summary statistics of the results obtained from the SIRS model for the coinfection strain Deltacron across different network structures.**

| Quantity | Description | WS(P = 0) | WS(P = 0.1) | WS(P = 1) | ER | BA |
|---|---|---|---|---|---|---|
| $I_{1max}$ | Peak domain of dynamics of infected individuals to strain 1 | 125 | 409 | 416 | 385 | 398 |
| $I_{2max}$ | Peak domain of dynamics of infected individuals to strain 2 | 324 | 3006 | 3098 | 3208 | 3167 |
| $I_{12max}$ | Peak domain of dynamics of co-infected individuals to strain 12 | 26 | 506 | 1269 | 1177 | 1159 |
| $I_{1avg}$ | The total mean of dynamics of infected individuals to strain1 | 17 | 89 | 118 | 108 | 112 |
| $I_{2avg}$ | The total mean of dynamics of infected individuals to strain 2 | 77 | 636 | 584 | 598 | 592 |
| $I_{12avg}$ | The total mean of dynamics of co-infected individuals to strain 12 | 2 | 100 | 235 | 218 | 215 |
| $S_{min}$ | The minimum value of dynamics of susceptible individuals at the peak of the disease | 9452 | 5923 | 5178 | 5232 | 5240 |
| $t_{max}(\delta)$ | The time of culminated dynamics of infected individuals to Delta strain | 44 | 42 | 51 | 51 | 51 |
| $t_{max}(o)$ | The time of culminated dynamics of infected individuals to Omicron strain | 76 | 54 | 31 | 32 | 32 |
| $t_{max}(Deltacron)$ | The time of culminated dynamics of infected individuals to Deltacron | 35 | 68 | 48 | 49 | 49 |
| *TotalInfected* | The total stats of infected individuals to disease | 3414 | 13085 | 17258 | 16737 | 16631 |
| *TotalRecovered* | The total stats of recovered individuals from disease | 3172 | 11907 | 15075 | 14694 | 14606 |
| *TotalRemovedduetoinfection* | The total stats of removed individuals from disease | 201 | 843 | 1385 | 1304 | 1301 |

In accordance with Fig 3d, the Delta strain, known for its high severity and wide range of diseases and mortality, diminishes from the epidemic scene when the Omicron strain is present. This disappearance of the dangerous Delta and Deltacron strains occurs within approximately 250 days, resulting in a reduced disease prevalence.

The abundance of infected individuals to the Delta and Omicron strains for their first, second, third and fourth occurrences is displayed in Fig 11. As the Omicron strain exhibits dominance, the number of engagements with this strain surpasses that of the Delta strain and spans a wider range. In the WS network with a rewiring probability of P = 0.1 (small-world feature), the prevalence domain of the Delta strain is nearly equivalent to that of the random networks for the first occurrence of infection, while it doubles for the second occurrence. This difference can be attributed to the substantial clustering observed in the small-world network.

In the coinfection model (refer to Fig 12), the Omicron strain exhibits progression up to the third occurrence and during the second and third occurrences, the WS network with a rewiring probability of P = 0.1 generates a larger domain compared to the random networks. Moreover, the prevalence of the coinfection state in this network is significantly lower than in random networks. This discrepancy arises from the fact that networks with a small-world feature exhibit a higher average ratio of clustering coefficient to average path length compared to random networks. Consequently, numerous clusters form within the small-world network, resulting in a higher prevalence of single infections within these clusters. As a result, individuals infected with one strain are less likely to encounter the second strain within their local neighborhood, leading to a lower incidence of coinfection. In random networks, however, the distribution of both strains is more random throughout the network, promoting a more

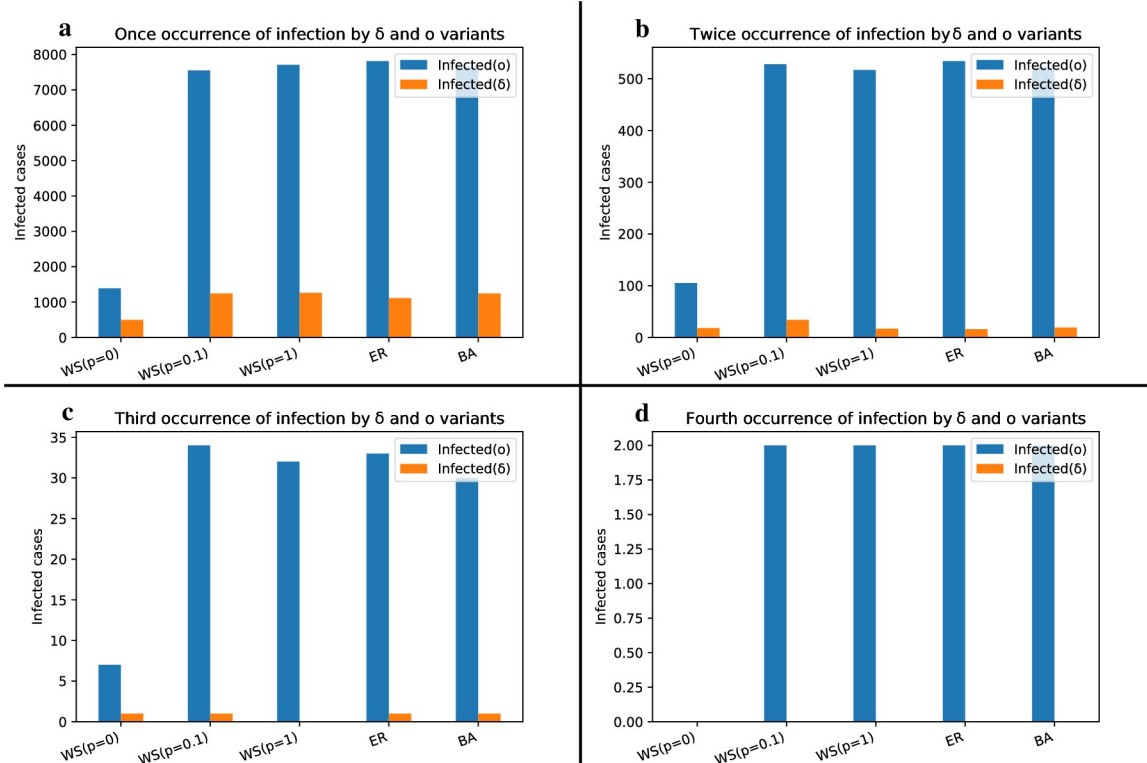

**Fig 11. Bar plot depicting the cumulative count of individuals experiencing multiple infections by the Delta and Omicron strains in the competitive model.**

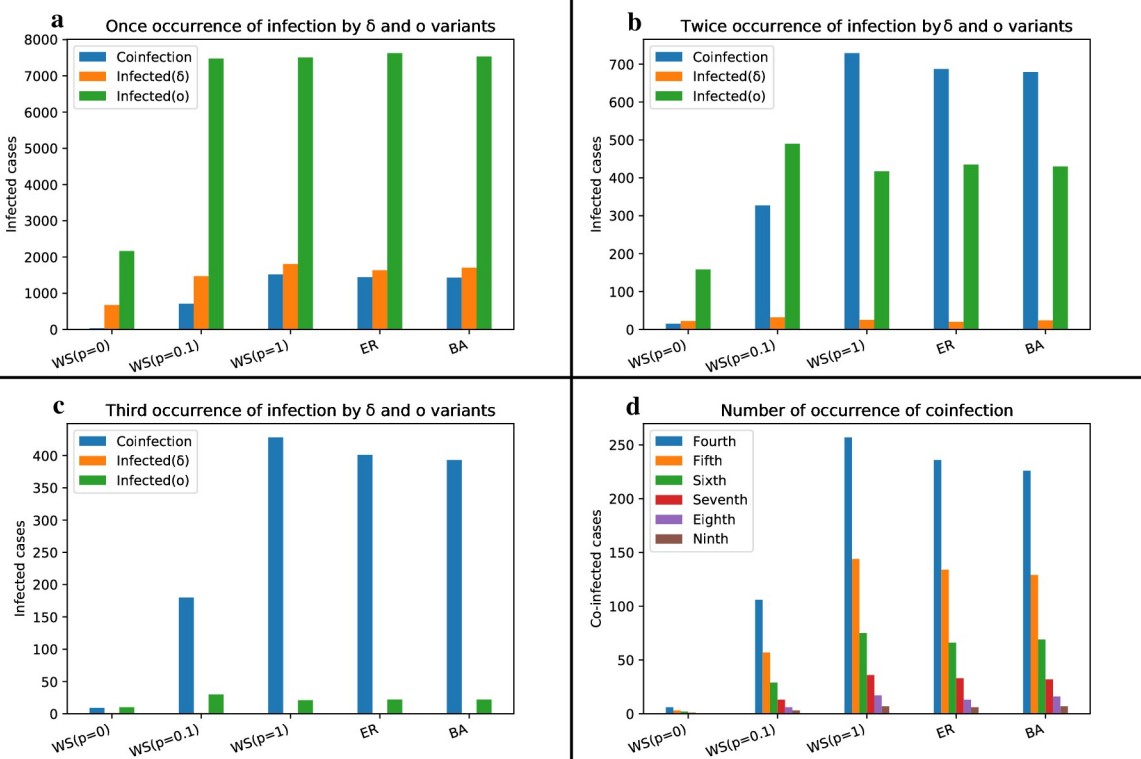

**Fig 12. Bar chart illustrating the cumulative count of individuals infected multiple times by the Delta, Omicron and Deltacron strains in the coinfection model.**

widespread occurrence of coinfection in the homogeneous Erdős-Rényi network (characterized by greater randomness) and the heterogeneous Barabasi-Albert network (featuring the presence of hubs).

## Conclusion

This study comprehensively investigated two epidemic models, focusing on single infection and coinfection scenarios to examine the dynamics of multi-infectious models. The Alpha and Beta strains, characterized by higher transmissibility, contributed to a larger epidemic range. Moreover, the Delta strain emerged as significantly more virulent and invasive than previous strains, warranting major concern.

Among the Variants of Concern (Omicron, Alpha, Beta, Gamma and Delta), the relatively less virulent Omicron strain assumed a dominant role and became the primary focus of investigation. Its interactions with other Variants of Concern were explored across random networks and various social network structures. In each case, Omicron exhibited superior competitive advantage, leading to its widespread prevalence in society. By effectively displacing the more virulent strains, Omicron prevented further transmission of infections and facilitated the transition of infected individuals into the removed class.

This dominance of the Omicron strain is prominently evident in the delayed state, where its introduction as the secondary strain gradually diminished the prevalence of more virulent strains. Remarkably, Omicron acted akin to a vaccine by swiftly suppressing the dangerous strain within the population.

Furthermore, in the coinfection state within the Erdős-Rényi (ER) network, an increase in the coinfection coefficient (a) corresponded to an escalation in the dynamics of coinfection individuals, while the dynamics of infected individuals to the Delta strain diminished over time. When analyzing the impact of network topology, it became apparent that the high ratio of average clustering coefficient to average path length in the Watts Strogatz (WS) network with small-world features, in comparison to random and Barabasi-Albert (BA) networks, resulted in a reduced occurrence of coinfection. This can be attributed to the formation of clusters predominantly hosting single infections and having fewer neighboring individuals infected with the second strain within their adjacency.

In summary, this research sheds light on the dynamics of epidemic models, highlighting the dominance of the Omicron strain as a less virulent yet highly competitive variant. The findings underscore the efficacy of Omicron in supplanting more virulent strains and its potential role in mitigating the spread of infectious diseases. The study also emphasizes the influence of network topology on the occurrence of coinfection, with the small-world structure exhibiting distinct characteristics that limit the prevalence of simultaneous infections. These insights contribute to our understanding of epidemic dynamics and can inform strategies for disease control and prevention.

## Supporting information

**S1 Data.**
(ZIP)

## Author Contributions

**Conceptualization:** Hamed Jabraeilian, Yousef Jamali.

**Data curation:** Hamed Jabraeilian.

**Formal analysis:** Hamed Jabraeilian.

**Funding acquisition:** Hamed Jabraeilian.

**Investigation:** Hamed Jabraeilian.

**Methodology:** Hamed Jabraeilian, Yousef Jamali.

**Project administration:** Yousef Jamali.

**Supervision:** Yousef Jamali.

**Validation:** Yousef Jamali.

**Visualization:** Yousef Jamali.

**Writing – original draft:** Hamed Jabraeilian.

**Writing – review & editing:** Yousef Jamali.

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
