## [Decision Letter · Decision Letter 0]

9 Aug 2023

Omicron vs. the Rest: Assessing the Competitive Dynamics and Coinfection Scenarios of COVID-19 Strains on a Social Network

PONE-D-23-18040

Dear Dr. Jamali,

We’re pleased to inform you that your manuscript has been judged scientifically suitable for publication and will be formally accepted for publication once it meets all outstanding technical requirements.

Kind regards,

Junyuan Yang

Academic Editor

PLOS ONE

Journal Requirements:

1.** **Please update your submission to use the PLOS LaTeX template. The template and more information on our requirements for LaTeX submissions can be found at http://journals.plos.org/plosone/s/latex.

Additional Editor Comments (optional):

Reviewers' comments:

Reviewer's Responses to Questions

**Comments to the Author**

1. Is the manuscript technically sound, and do the data support the conclusions?

Reviewer #1: Yes

2. Has the statistical analysis been performed appropriately and rigorously? 

Reviewer #1: Yes

3. Have the authors made all data underlying the findings in their manuscript fully available?

Reviewer #1: Yes

4. Is the manuscript presented in an intelligible fashion and written in standard English?

Reviewer #1: Yes

5. Review Comments to the Author

Reviewer #1: This paper use a SIRS infectious disease on network model explains the competition, coexistence and evolution between multiple strain variants of Covid-19, it contributes to the explanation of the infection dynamics of the novel coronavirus.

6. PLOS authors have the option to publish the peer review history of their article (what does this mean?). If published, this will include your full peer review and any attached files.

Reviewer #1: No

---

## [Editor Report · Acceptance letter]

19 Sep 2023

PONE-D-23-18040 

Omicron vs. the Rest: Assessing the Competitive Dynamics and Coinfection Scenarios of COVID-19 Strains on a Social Network 

Dear Dr. Jamali:

I'm pleased to inform you that your manuscript has been deemed suitable for publication in PLOS ONE. Congratulations! Your manuscript is now with our production department. 

Kind regards, 

on behalf of

Dr. Junyuan Yang 

Academic Editor

PLOS ONE